# Hydrogel-Based Pre-Clinical Evaluation of Repurposed FDA-Approved Drugs for AML

**DOI:** 10.3390/ijms24044235

**Published:** 2023-02-20

**Authors:** Jenna R. James, Johnathan Curd, Jennifer C. Ashworth, Mays Abuhantash, Martin Grundy, Claire H. Seedhouse, Kenton P. Arkill, Amanda J. Wright, Catherine L. R. Merry, Alexander Thompson

**Affiliations:** 1Blood Cancer and Stem Cell Group, School of Medicine, Biodiscovery Institute, University of Nottingham, Nottingham NG7 2RD, UK; 2Optics and Photonics Research Group, Faculty of Engineering, University of Nottingham, Nottingham NG7 2RD, UK; 3Stem Cell Glycobiology Group, School of Medicine, Biodiscovery Institute, University of Nottingham, Nottingham NG7 2RD, UK; 4School of Veterinary Medicine & Science, University of Nottingham, Nottingham NG7 2RD, UK; 5Endothelial and Vascular Imaging Laboratories, Biodiscovery Institute, University of Nottingham, Nottingham NG7 2RD, UK

**Keywords:** acute myeloid leukemia, 3D peptide hydrogel model, candidate drugs

## Abstract

In vivo models of acute myeloid leukemia (AML) are low throughput, and standard liquid culture models fail to recapitulate the mechanical and biochemical properties of the extracellular matrix-rich protective bone marrow niche that contributes to drug resistance. Candidate drug discovery in AML requires advanced synthetic platforms to improve our understanding of the impact of mechanical cues on drug sensitivity in AML. By use of a synthetic, self-assembling peptide hydrogel (SAPH) of modifiable stiffness and composition, a 3D model of the bone marrow niche to screen repurposed FDA-approved drugs has been developed and utilized. AML cell proliferation was dependent on SAPH stiffness, which was optimized to facilitate colony growth. Three candidate FDA-approved drugs were initially screened against the THP-1 cell line and mAF9 primary cells in liquid culture, and EC50 values were used to inform drug sensitivity assays in the peptide hydrogel models. Salinomycin demonstrated efficacy in both an ‘early-stage’ model in which treatment was added shortly after initiation of AML cell encapsulation, and an ‘established’ model in which time-encapsulated cells had started to form colonies. Sensitivity to Vidofludimus treatment was not observed in the hydrogel models, and Atorvastatin demonstrated increased sensitivity in the ‘established’ compared to the ‘early-stage’ model. AML patient samples were equally sensitive to Salinomycin in the 3D hydrogels and partially sensitive to Atorvastatin. Together, this confirms that AML cell sensitivity is drug- and context-specific and that advanced synthetic platforms for higher throughput are valuable tools for pre-clinical evaluation of candidate anti-AML drugs.

## 1. Introduction

Acute myelogenous leukemia (AML) is the most common type of acute leukemia in adults and arises from genetic mutations in hematopoietic stem or progenitor cells, which result in an uncontrollable proliferation of immature leukemic white blood cells and a blockage of differentiation [1,2]. AML is an aggressive, heterogeneous disease that can present in all age groups with an overall poor prognosis [3]. Increasing age correlates with incidence, with the highest rates observed in people over 75 [4,5]. Due to the intensity of standard chemotherapies, the cure rate for patients older than 60 years is only 5–15% [6]. A better understanding of key pathways, molecular classification, and monitoring of patients over the last two decades has resulted in the discovery of promising novel therapies, including FLT-3 inhibitors and Venetoclax (reviewed by Bazinet and Assouline [7]).

An increasingly important challenge in cancer drug development is the cost-benefit ratio. As of 2020, developing a drug treatment from discovery to clinical application costs an average of 2.5 billion dollars, with the most expensive part of the process being clinical trials, which consume approximately 50% of the total investment [8,9]. Drug repurposing—a system for identifying new applications for drugs outside the disease of their initial approval [10,11,12]—is emerging as a potential strategy for future cancer treatments. FDA-approved repurposed drugs have long-term toxicity profiles that minimize the risk of failure in the clinic due to side effects and thus reduce the overall time and cost. The major effort is determining the efficacy of the repurposed drugs for the specific disease of interest. We and others have demonstrated the efficacy of several drugs targeted towards other diseases (e.g., sodium valproate) in the clinical setting against subtypes of AML [9,13]. More recently, using a combination of connectivity mapping in a conditional model of Mixed-Lineage Leukemia (MLL) AML and drug similarity analysis, we identified FDA-approved candidate drugs (including Salinomycin, Atorvastatin, and Vidofludimus) that potentially target *HOXA*-dependency in MLL-AF9 leukemia [14]. For candidate drug studies, the human THP-1 cell line and murine mAF9 primary cells were selected as matched exemplars of AML models that harbor the *MLL-AF9* translocation. 

Due to their ability to recapitulate the bone marrow niche, animal models remain the current gold standard for pre-clinical AML research. However, the combination of difficulty in interpreting key parameters that guide cell behavior [15,16], and the lack of translation of positive findings to humans [17], suggests an inability of current preclinical testing to accurately mimic the AML microenvironment and highlights the need for the development of defined, customizable models. The structure-function relationship within 3D models has been shown to significantly impact how cellular forces are generated and transduced into biochemical or structural changes [18]. The addition of a 3D support helps regulate a range of tissue mechanics, including the transport of nutrients and signaling molecules, cell cytoskeleton organization, and cell morphology [18,19]. Compared to 2D models, this behavior within 3D models has been shown to be more representative of the in vivo setting from a biophysical perspective, and as such, there is a great need for accessible and tunable 3D platforms to mimic mechanical cues within AML disease models [20,21,22].

Hydrogels can be created using natural or synthetic components and prepared by various methods depending on the desired functionality and application [23]. They have emerged as 3D surrogate platforms for various disease models and applications, including scaffolds, drug delivery, and cell transplantation, to mimic a wide range of tissues such as neural tissue, breast tissue, or bone [23,24,25,26,27,28]. Compared to other methods for generating 3D models, e.g., the hanging-drop method or a bioreactor to drive cells to self-aggregate, hydrogels have the advantage of being able to provide both chemical and mechanical cues to resident cells. This is particularly important when modelling the AML microenvironment, as matrix stiffness has previously been shown to directly affect leukemic cell proliferation and resistance to standard chemotherapy [22]. Due to their biological origin, commonly used platforms, including Matrigel^®^ and, to a lesser extent, collagen or hyaluronan hydrogels, suffer from the fact that they are compositionally undefined and susceptible to batch-to-batch variation, particularly around matrix stiffness. Conversely, synthetic hydrogels are typically well-defined, have limited batch-to-batch variability, and often provide the end user with the ability to tune their chemical and mechanical properties.

Herein, we use an unmodified, biologically inert synthetic self-assembling octapeptide hydrogel (SAPH) previously optimized to create supportive environments for a variety of cell types [29], with the ability to independently modify biochemical composition and subsequent functionalization by addition of relevant extracellular matrix (ECM) molecules, proteins, and growth factors [30,31]. The SAPH hydrogel provided a ‘blank canvas’ platform capable of maintaining growth and monitoring of leukemic cell lines, primary leukemia and patient samples. Candidate repurposed drug treatments of leukemia cells in 3D hydrogels were compared to current liquid and methylcellulose-based platforms.

## 2. Results

### 2.1. Unmodified SAPH Hydrogels Provide a Platform for Human Leukemia Cell Line Maintenance and Survival

For optimization of culture methods within the SAPHs, several subtypes of human leukemic cell lines were studied within the unmodified 3D model; these included U937, THP-1, MV4-11, HL60, and OCI-AML3. The prototypical U937 suspension cell line (average diameter 14 µm) was used to optimize the concentration of peptide in the hydrogel that best supported AML cell survival and growth. A concentration of 6 mg/mL was identified as optimal for U937 cells (Figure 1A) and used as the basis for other cell types. All human leukemic cells survived and proliferated within the SAPH with some variation in growth dynamics between cell lines (Figure 1B). The proliferation of the cells varied in colony size and density, ranging from smaller, more compact colonies, as seen with THP-1 and HL60 cells, to larger, broader colonies as seen in the U937 and OCI-AML-3 cell cultures. The optical transparency of the SAPHs allows for quantitative readouts of cell proliferation. Seeding density does not appear to affect colony size in any cell line investigated, with both cell concentrations studied producing similar-sized colonies. This supported the hypothesis that the colonies are formed by cell proliferation rather than by cell migration from the local microenvironment.

### 2.2. Maintenance and Survival of Primary Murine Leukemia Cells in Unmodified and Modified SAPH

mAF9 cells were seeded at different concentrations in standard (1×) or reduced growth factor (0.5×) liquid culture media in unmodified (top panel) or modified (bottom panel) hydrogels (Figure 2). Phase-contrast images taken 3 or 6 days following the initial cell seeding clearly demonstrate colony formation of mAF9 cells in both unmodified and modified SAPH. Seeding densities of 2 × 10^5^ in reduced growth factor media (Condition 2) or 1 × 10^5^ in standard growth factor media (Condition 4) produced a quantifiable level of colony growth formation, according to standard counting techniques. Staining and confocal imaging of the colonies (Calcein AM and Ethidium homodimer-1), taken 6 days following initial seeding, supported the phase-contrast results and demonstrated little or no cell death (red cells) by day 6 in mAF9 colony growth. The addition of methylcellulose (Condition 5), laminin (Condition 6), fibronectin (Condition 7), or hyaluronan (Condition 8) increased the number of mAF9 colonies produced from the standard cell seeding and growth factor media concentrations used in Condition 1.

### 2.3. Salinomycin, Vidofludimus, and Atorvastatin Show Efficacy in Liquid Culture Treatment of Human and Murine AML Cells

To initially evaluate the potential efficacy of the candidate drugs, a broad range of dosages of Salinomycin, Vidofludimus, and Atorvastatin were completed within liquid culture for the THP-1 human leukemic cell line (Figure 3A) and the mAF9 primary mouse leukemic cells (Figure 4A). Cells were assessed for viability at 24, 48, and 72 h following treatment with each drug of interest and recorded as percentage survival compared to vehicle control. The EC50 values, calculated where possible (Figure 3B and Figure 4B), were in the micromolar or sub-micromolar range for all drugs in both models by 72 h. Short-term drug treatment (24 h) of mAF9 cells showed measurable efficacy that was not seen in the growth factor-independent THP-1 cells, indicating greater sensitivity for the primary cells. Atorvastatin and Salinomycin both showed time- and dose-dependent reductions in cell viability across both models. Low dose Salinomycin (<250 nM) was sufficient to cause a 50% reduction in cell viability in mAF9 cells for all timepoints evaluated.

### 2.4. Salinomycin and Atorvastatin Demonstrate Drug Efficacy in THP-1 Cells Enapsulated in SAPH

To evaluate the potential efficacy of candidate drugs in 3D culture, the calculated EC50 values for Salinomycin, Vidofludimus, and Atorvastatin for THP-1 cells in liquid culture were used for encapsulated SAPH models. An ‘early-stage’ model, to mimic leukemia initiation, was generated by treating hydrogels with the candidate drug or vehicle control during cell encapsulation (~10 min). To evaluate the effect of drug treatment on previously formed leukemia, an ‘established’ model was generated whereby drug or vehicle were added 72 h following complete THP-1 cell encapsulation within the SAPHs. In support of the liquid culture results, Salinomycin and Atorvastatin showed efficacy against THP-1 leukemia cells in both the ‘early-stage’ (Figure 5A) and ‘established’ (Figure 5B) encapsulated 3D SAPH models. The sensitivity to Salinomycin treatment in SAPHs was comparable to that observed in liquid culture of THP-1 cells for all timepoints. Short-term (24 h) Atorvastatin treatment demonstrated reduced sensitivity in both ‘early-stage’ and ‘established’ SAPH models. THP-1-encapsulated cells demonstrated resistance to Vidofludimus using the established liquid culture EC50 values.

### 2.5. Decreased Viability and Colony Formation in Salinomycin and Atorvastatin-Treated AML Patient Samples

To further examine the efficacy of Salinomycin and Atorvastatin, the two drugs that demonstrated consistent effects on THP-1 and mAF9 cells, they were applied to AML patient samples. As liquid culture of primary AML patient samples alone is problematic, cell viability assays for 3D SAPH models were compared with an established liquid culture plus MS-5 stromal cell co-culture (Figure 6A). As can be seen, Atorvastatin treatments in the co-culture setting produced ~50% reduction in cell viability in both patient samples following 72 h treatment, whereas in this setting, Salinomycin only showed a reduction (~40%) in survival for patient #694. In the 3D SAPH cultures, Salinomycin showed a consistent time- and dose-dependent reduction in cell viability for both AML patient samples (#522 and #694 denoted by *). Longer-term and functional effects of the drugs were also monitored using methylcellulose colony assay formation (Figure 6B). Salinomycin (500 nM) was more potent than Atorvastatin (5 µM) in reducing colony numbers (right panel) and the colonies that were produced following either drug treatment were of the more differentiated type II subtype than vehicle control treatment (left panel). 

## 3. Discussion

The development of pathophysiologically relevant in vitro cancer models for drug discovery is one of the major challenges facing researchers in the desire to improve the low translation of therapies into clinics [8,32]. This study presented for the first time a defined SAPH model free of matrix motifs to examine the impact of 3D structure on AML cell sensitivity to drug treatments. This novel approach enabled direct comparison of the AML treatment response of repurposed drugs in liquid culture and simple, non-functionalized 3D SAPH, providing unique insight into the importance of mechanical cues within AML models.

Increased peptide concentration, resulting in a stiffer gel (Appendix A), affected both the number of cell colonies formed and the structure of the colonies. As the concentration increased from 6 mg/mL to 10 mg/mL, the SAPH continued to support cell proliferation; however, the colonies became fewer in number and appeared denser and more rounded (Appendix A). 

Limited cell proliferation was observed within the maximum stiffness SAPH investigated (15 mg/mL). This concentration equates to a storage modulus of ~5 kPa, whereas 6 mg/mL SAPH yields a storage modulus of ~600 Pa as measured by bulk rheology [33]. Maximum leukemia cell proliferation over 72 h occurred in 6 mg/mL SAPH, which was higher than established liquid culture. For this reason, 6 mg/mL SAPH was used for all future leukemia cell experiments. Both the THP-1 human AML cell line and mAF9 primary leukemias grew well in 6 mg/mL SAPH following optimized seeding densities which provided a basis for the drug treatments.

Dihydroorotate dehydrogenase (DHODH) is a key enzyme that catalyzes the rate-limiting step in pyrimidine biosynthesis. Vidofludimus is a DHODH inhibitor with potential anti-inflammatory, immunomodulating, and anti-viral activities, recently shown to promote cell cycle arrest in lymphoblastoid and lymphoma cell lines [34]. Although the MLL-AF9 harboring THP-1 and mAF9 cells showed limited sensitivity to Vidofludimus, recent reports suggest that second-generation DHODH inhibitors may have a particular role to play in MLL-AF9 AML and are now available at low concentrations (74 nM) with low toxicity [35].

Targeting metabolic pathways is becoming an established anti-cancer strategy. Anti-cholesterol drugs such as statins are reported to inhibit the proliferation and survival of cancer cells, including leukemia, alone or in combination with other drugs [36]. Atorvastatin in particular, as a single agent, has been shown to induce cell cycle arrest, induce apoptosis, and inhibit the YAP pathway in HL60 and K562 cells [37].

Salinomycin is a coccidiostat ionophore first identified as an antibacterial drug. In a screen of over 16,000 compounds, Salinomycin was found to be one hundred times more effective than paclitaxel in the treatment of breast cancer in mice with selectivity against cancer stem cells [38]. Following this, Roulston et al. showed that Salinomycin could eradicate human AML cell lines and mouse primary AML cells in liquid culture without affecting the colony formation of normal hematopoietic cells [39]. Salinomycin is reported to have anti-cancer effects by accumulating and sequestering iron in lysosomes [40] leading to local iron deficiency. In response to this, target cells degrade ferritin [41] to restore local iron levels, which leads to increased production of reactive oxygen species, lysosomal membrane permeabilization, and cell death consistent with ferroptosis. As cancer stem cells are characterized by high intracellular iron content, Salinomycin is able to selectively target them [42].

Vidofludimus was the only candidate drug to show consistently decreased drug sensitivity in the SAPH models compared to liquid culture. Decreased drug sensitivity in the hydrogel setting may be due to reduced perfusion in the 3D models due to several factors, including size, structure, and charge of the drug. However, reduced perfusion is unlikely as normal leukemia cell proliferation and growth throughout the 3D cultures indicate that the full depth of the gel is perfused by the media components, including large and charged serum proteins. Another possible explanation for decreased drug sensitivity is impairment of local accessibility due to cell encapsulation. Ishikawa et al. demonstrated that the tumour microenvironment can induce dormancy in leukemia cells, potentially increasing the resistance to anti-cancer drugs [43]. Encapsulation of cells in the SAPH and induced ECM formation by the cells could act as a mechanical cue and create a protective leukemia microenvironment [44,45,46]. This could explain, in part, the reduced sensitivity to Atorvastatin in the ‘early stage’ SAPH model compared to the ‘established’ model, but would require further analysis.

Of the three candidate drugs investigated, Salinomycin and Atorvastatin demonstrated more reliable and measurable anti-leukemia effects across all pre-clinical models used. The data presented for Salinomycin supports and extends our previous findings [39] to include the MLL-AF9-expressing THP-1 cell line along with the 3D SAPH platform. Within this study, Salinomycin was the only drug to exhibit a similar EC50 value in all three models investigated (liquid culture (2D), ‘early-stage’ gel, and ‘established’ gel). This indicates that the 3D structure of the SAPH, or encapsulation of cells, did not have a significant negative biological impact on the cells’ responsiveness to Salinomycin. However, the reduced sensitivity to Atorvastatin in the ‘early stage’ model, as cells are becoming encapsulated, suggests that impaired biological response may be drug and cell context dependent in our AML-SAPH model, as reported for other cell types and hydrogel systems [15,17,19,47].

Candidate drug sensitivity studies on primary AML patient samples are notoriously difficult due to poor ex vivo growth caused by spontaneous differentiation and increased cell death. Co-culturing on MS-5 stromal cells is an established protocol to improve the viability of AML blasts in liquid culture [48]. Two AML patient samples were able to be maintained in the co-culture and SAPH models. Salinomycin effectively eradicated the AML cells from both patients within three days of treatment in the SAPH model. However, this response was blunted in the co-culture model, suggesting that additional survival cues from stromal cells impaired the mechanism of action of Salinomycin. Atorvastatin proved more successful in eliminating AML cells in the co-culture setting compared to the SAPH model, supporting reports that the YAP pathway and focal adhesion are inhibited by this molecule.

## 4. Materials and Methods

### 4.1. Cell Culture

Human leukemic cell lines (THP-1, U937, MV4-11, HL60, and OCI-AML3) were maintained in suspension culture in RPMI-1640 supplemented with 10% fetal bovine serum (FBS) and 1% L-Glutamine (all supplied by Gibco, Life Technologies, Waltham, MA, USA) and incubated at 37 °C and 5% CO_2_ in a humidified atmosphere. Cultures were passaged every 2–3 days to maintain a cell number of 1 × 10^5^/mL to 1 × 10^6^/mL. Primary murine cells (mAF9) were previously expanded from MLL-AF9 leukemic mice generated by retroviral transfection of donor primary bone marrow cells. For continuous culture of MLL-AF9 mouse primary leukemic cells (mAF9) the cells were cultured in leukemic mouse media i.e., RPMI-1640 with additions of 10% FBS (HyClone, Logan, UT, USA), 1% L Glutamine, 5 ng/mL mIL-3 (#130-099-508), 5 ng/mL mIL-6 (#130-096-682), 5 ng/mL mGM-CSF (#130-095-742), 50 ng/mL mSCF (130-101-693; all growth factors from Miltenyl Biotec, Bisley, UK). Phase-contrast or viability dye-based imaging (Calcein AM and ethidium homodimer) was used to periodically observe cell growth and viability (Eclipse TI-S; Nikon, Tokyo, Japan).

### 4.2. SAPH Preparation

Peptides (SAPH, Phe-Glu-Phe-Glu-Phe-Lys-Phe-Lys, Cambridge Research Biochemicals, UK/Pepceuticals, Leicester, UK) at the desired concentration were dissolved in sterile water (W3500, Sigma Aldrich, St. Louis, MO, USA). Following incubation at 80 °C for 2 h, 0.5 M NaOH was used to elevate the hydrogel pH until an optically clear, self-supporting, viscous solution was formed. 10× phosphate buffered saline (PBS) (Gibco, Life Technologies, Waltham, MA, USA) was added to produce a final hydrogel of 1× PBS. Hydrogels were vortexed and centrifuged at 1000 rpm (Heraeus Megafuge 40R, Thermo Scientific, Waltham, MA, USA) after each addition to ensure a homogeneous composition. Following all additions, the hydrogels were incubated overnight at 80 °C and then stored at 4 °C until required for cell encapsulation. All batches of peptide were analyzed for similarity in rheological properties prior to use to ensure comparability of results. Briefly, SAPH preparations were mounted onto a Physica MCR 301 rheometer (Anton Paar, Graz, Austria) with the Peltier plate set to 37 °C and the linear viscoelastic region for each set of samples determined. Firstly, an amplitude sweep from 0.1 to 100% strain at 1 rad/s was done to determine a range that would not destroy the structure of the sample. A frequency sweep with a constant strain of 0.5% and constant frequency of 1 rad/s, with 10 measurements in 5 min, was then used to determine the approximate stiffness, expressed as its Young’s modulus (Appendix A).

### 4.3. Cell Encapsulation into Unmodified SAPHs

Hydrogels were incubated at 80 °C for at least 2 h prior to encapsulation to decrease viscosity, then placed into a 37 °C water bath during cell preparation. Suspension cells were counted, re-suspended at required concentrations, and 250 μL was carefully mixed into 1 mL of hydrogel, creating a media ratio of 1:5 and a final cell concentration of 1 × 10^4^/mL to 1 × 10^5^/mL for human cell lines and patient samples, and 5 × 10^4^/mL to 5 × 10^5^/mL for mAF9 cells. The cell-seeded SAPH were then transferred into 8-well µ-slides (Ibidi, Germany) or 24-well, PET 1 μm, transwell plates (Millicell Cell Culture Inserts, Millipore, Burlington, MA, USA), or 96-well plates (ViewPlate-96 Black, Perkin Elmer, Waltham, MA, USA) and incubated at 37 °C/5% CO_2_ in a humidified atmosphere for 10 min (Appendix A). Appropriate cell media was then added on top of the SAPH, and at least two media changes were performed in the following hour to fully neutralize the SAPH. Media changes were subsequently performed every 1–3 days, depending on the experiment.

For the ‘early-stage’ drug treatment leukemia models, following 10 min of cell incubation with SAPH, 200 µL of the relevant media was added on top of the gel. Following two media changes within an hour after seeding, 100 µL of media with RealTime-Glo™ (Promega, Chilworth, UK) and the required drug/DMSO control were added to the SAPH. For the ‘established’ models, encapsulated cells were cultured without vehicles or drugs for 72 h but with a media change after 24 h. At 72 h, the RTG and required drug concentrations were added in the same way as described for the ‘early-stage’ model. All plates were protected from light and incubated at 37 °C with 5% CO_2_ in a humidified atmosphere. Luminescence plate reads were performed at 24 h, 48 h, and 72 h.

### 4.4. Functionalization of 6 mg/mL SAPH

Modified SAPH incorporating matrix additions of methylcellulose, (MethocultTM GF M3434, Stem Cell Technologies, Vancouver, BC, Canada), laminin, (L2020, Sigma Aldrich, St. Louis, MO, USA), fibronectin, (07159, Stem Cell Technologies, Vancouver, BC, Canada), and hyaluronan, (HA1M, Lifecore Biomedical, Chaska, MN, USA) were created by preparing a 250 µL volume of cell suspension containing each matrix component at 5× the desired final concentration (diluted with 0.5× leukemic mouse media if required) and mixed with 1 mL of the precursor gel.

### 4.5. Cell Quantification Using Luminescence

Luminescence assays RealTime-Glo™ MT Cell Viability Assay (RTG) (Promega, Chilworth, UK) was routinely used to quantify the number of cells within liquid culture according to the manufacturer’s protocol. For 3D SAPH assays, 100 μL of cell-seeded gel was transferred to wells of a black 96-well plate in triplicates of each condition. Following the two required media changes after initial seeding of the plates, all media was removed and 100 μL of 2× RTG components diluted in media was added to produce a final concentration of 1× RTG (media and gel volume inclusive). The plates were protected from light and incubated at 37 °C and 5% CO_2_ in a humidified atmosphere for 1 h before luminescence plate reading (Fluorestar Omega Plate Reader, BMG LabTech, Ortenberg, Germany). Samples were prepared within black 96-well plates with an optically clear bottom to minimize background and bleed-through between wells. For all results collected using luminescence assays ‘no cell’ controls were performed to record any background fluorescence, which was averaged and removed from the results. All values were obtained in technical triplicates and were represented as a standard curve of known cell number against luminescence.

### 4.6. Drug Screening

Human leukemic cell lines and mAF9 cells were treated with a dose range of each candidate drug: Salinomycin (Sigma Aldrich, St. Louis, MO, USA), Vidofludimus, and Atorvastatin (both SelleckChem, Planegg, Germany). Drug treatments were compared to the well-established vehicle control (DMSO) at a concentration that does not affect cell survival (up to 0.2%) and is equal to the highest DMSO concentration within the drug dilutions. Graphing and the effective concentration of drug required to provoke a response halfway between the baseline and maximum responses (EC50 values) were calculated from non-linear regression analysis using Prism 8.3.1 statistical software (Graphpad, Boston, MA, USA).

### 4.7. AML Patient Samples

As a proof-of-principle, blood or bone marrow samples were obtained from AML patients #522 and #694 at diagnosis (Appendix A). Mononuclear cells were isolated using a standard density gradient centrifugation method using histopaque material and cryopreserved in liquid nitrogen. Only samples with >90% post-thaw viability were assayed. Samples at 1 × 10^6^/mL were treated as per cell lines in peptide hydrogels and cultured in RPMI supplemented with 10% FCS, 1% L-glut, 20 ng/mL IL3, 20 ng/mL SCF, 20 ng/mL IL-6, 25 ng/mL G-CSF (all R&D Systems, Oxfordshire, UK), and 0.07 µL/mL beta-mercaptoethanol with candidate drug or vehicle control (DMSO) in the presence or absence of MS-5 stromal cells (ACC441, DSMZ, Gottingen, Germany). Cell viability was assessed at 24, 48, and 72 h using the Real-Time Glo MT Cell viability assay (G9713, Promega, Chilworth, UK). Samples at 1 × 10^5^/mL were also cultured in methylcellulose (Methocult TM H4534, Stem Cell Technologies, Vancouver, BC, Canada) with candidate drug or vehicle control (DMSO). Colonies were counted by light microscopy on day 14. Graphing and analysis was performed using Prism 8.3.1 statistical software (GraphPad, Boston, MA, USA).

## 5. Conclusions

Together, the data highlights the effectiveness of unmodified SAPH with appropriate stiffness to support the cell growth of AML cells from different sources for drug sensitivity studies. Such models provide a robust platform to further modify and functionalize hydrogels to better mimic the hematopoietic niche, potentially for both leukemic and normal blood stem cell research. These advanced pre-clinical models will help identify and de-risk effective, non-toxic drugs for future clinical applications.

## Figures and Tables

**Figure 1 ijms-24-04235-f001:**
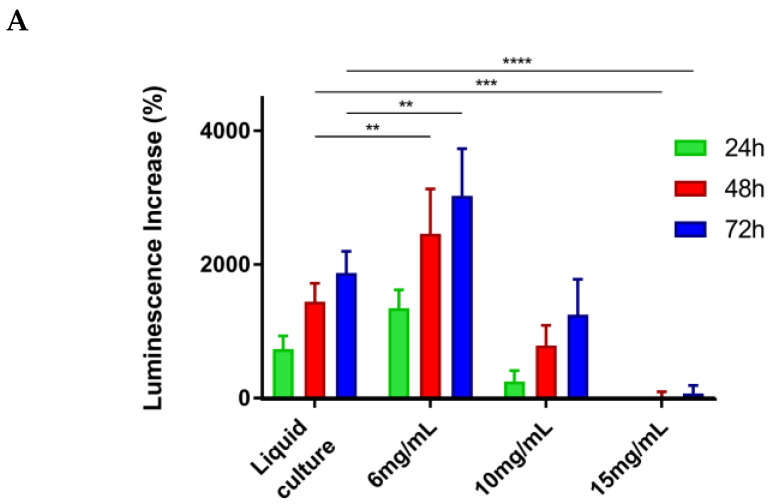
Human leukemia cell lines grow and form colonies in unmodified self-assembling peptide hydrogels (SAPH). (**A**) Cell viability assays determined by accrued luminescence measured by RealTime-Glo™; for U937 cells in increasing concentrations of hydrogel. Statistical significance (determined by two-way ANOVA in GraphPad Prism): ** *p* < 0.01, *** *p* < 0.001, **** *p* < 0.0001. N = 3, *n* = 3. (**B**) Phase-contrast images of human leukemic cell lines seeded at 5 × 10^4^/mL or 1 × 10^5^/mL in unmodified 6 mg/mL SAPH. Scale bars = 100 µm.

**Figure 2 ijms-24-04235-f002:**
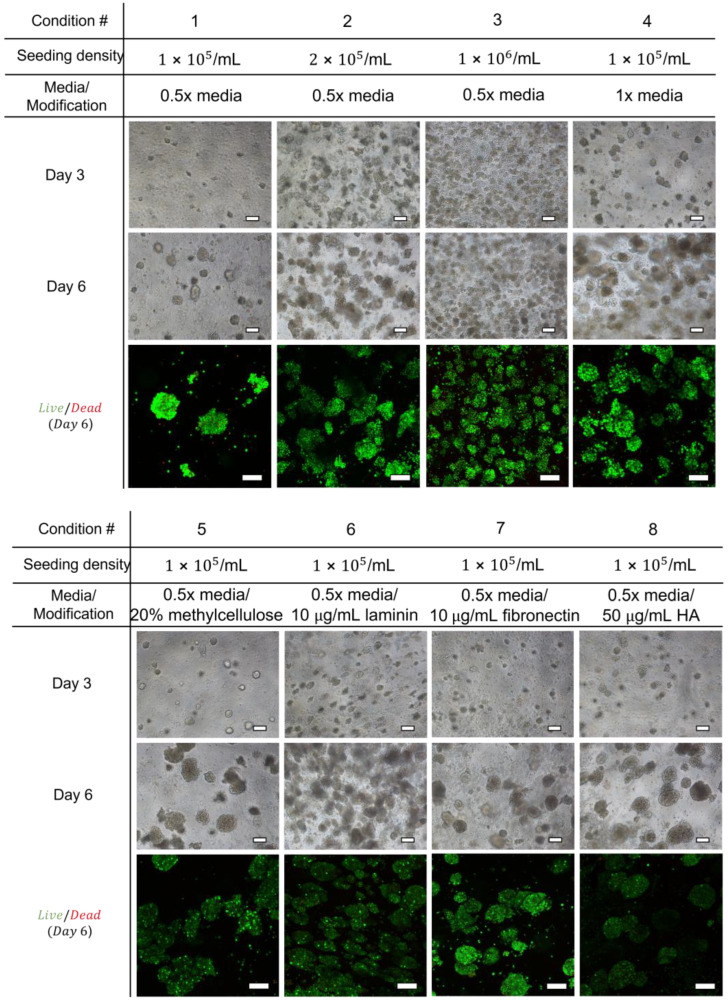
MLL-AF9 primary leukemia cells (mAF9) survive and form colonies in unmodified and modified self-assembling peptide hydrogels. Phase-contrast (top panel) and confocal images (bottom panel) of mAF9 cells in defined conditions of modified peptide hydrogels. Confocal images were taken 6 days following initial seeding and stained with Calcein AM and Ethidium Homodimer-1 to indicate viability. HA = Hyaluronan; scale bars = 100 µm.

**Figure 3 ijms-24-04235-f003:**
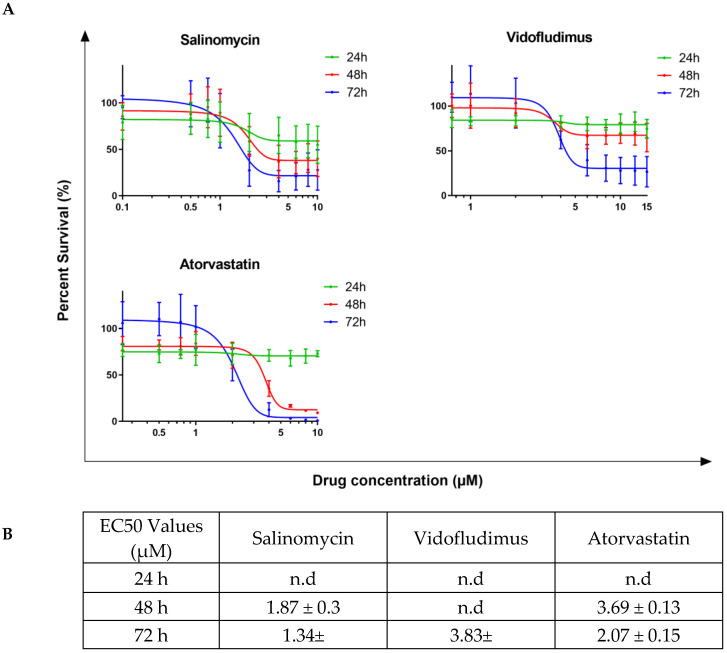
Candidate drug dose response assays for THP-1 cells in liquid culture. (**A**) Line graphs showing viability of THP-1 cells (as percentage survival) over time following candidate drug treatment at the indicated dosage as compared to 0.2% DMSO vehicle control. Mean values ± standard deviation of biological replicates (N = 4, *n* = 3) are plotted throughout. (**B**) Calculated EC50 values for each candidate drug at 24, 48, and 72 h following treatment.

**Figure 4 ijms-24-04235-f004:**
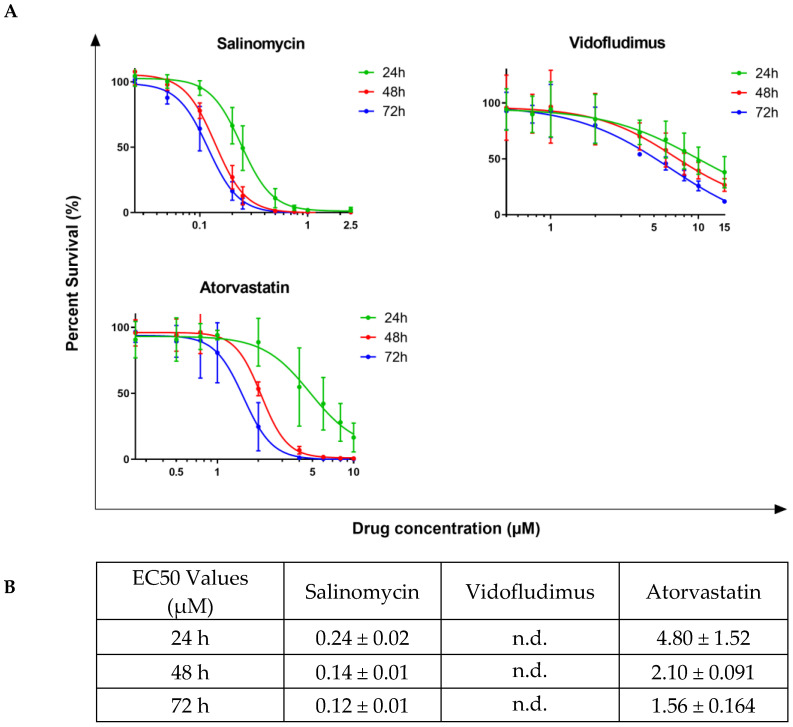
Candidate drug dose response assays for mAF9 cells in liquid culture. (**A**) Line graphs showing viability of mAF9 cells (as percentage survival) over time following drug treatment at the indicated dosage as compared to 0.2% DMSO vehicle control. Mean values ± standard deviation of biological replicates (N = 4, *n* = 3) are plotted throughout. (**B**) Calculated EC50 values for each candidate drug at 24, 48, and 72 h following treatment.

**Figure 5 ijms-24-04235-f005:**
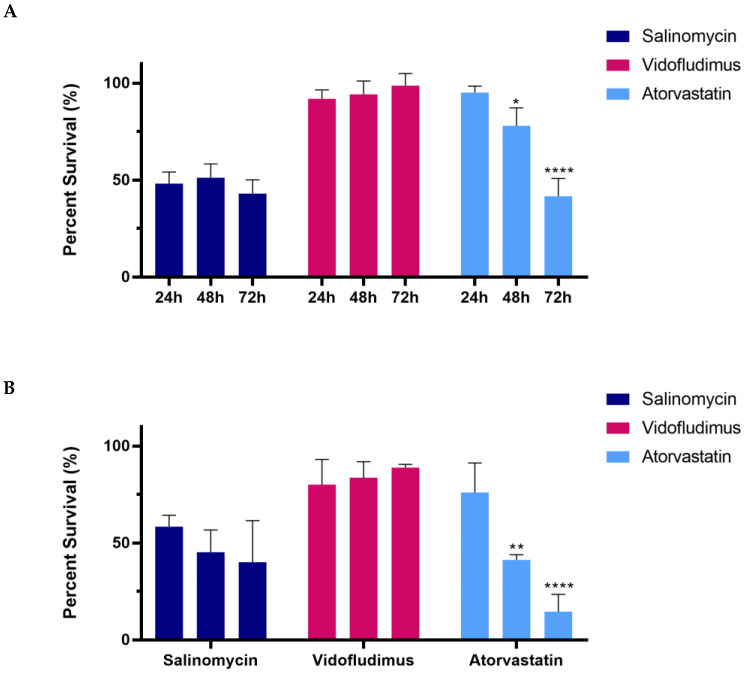
Candidate drug efficacy of THP-1 cells encapsulated in 6 mg/mL self-assembling peptide hydrogel (SAPH). (**A**) Bar graphs displaying the percentage survival of treated THP-1 cells 72 h after drug addition to ‘early-stage’ SAPH models. (**B**) Bar graphs displaying the percentage survival of treated THP-1 cells 72 h after drug addition in ‘established’ SAPH models. For all results, mean values ± standard deviation of biological replicates (N = 3, *n* = 3) are plotted throughout. Statistical significance (determined by two-way ANOVA in GraphPad Prism): * *p* < 0.05, ** *p* < 0.01, **** *p* < 0.0001. All results are presented as percentages compared to vehicle (DMSO) control of 0.02% for calculated EC50 values in liquid culture (Figure 3).

**Figure 6 ijms-24-04235-f006:**
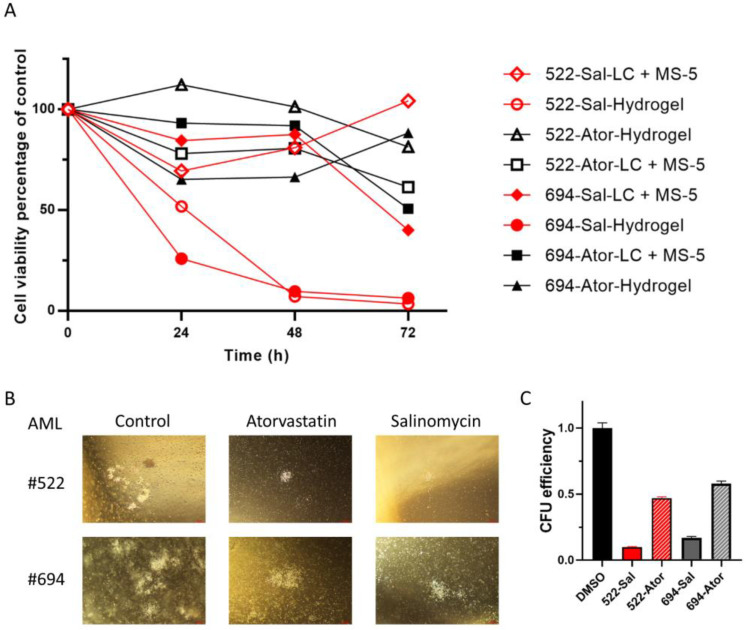
Atorvastatin and Salinomycin reduce the viability of AML patient samples in self-assembling peptide hydrogel (SAPH) culture and colony formation. (**A**) Line graphs displaying percentage survival of two AML patient samples (#522 and #694) treated with 5µM Atorvastatin (Ator) or 500 nM Salinomycin (Sal) for up to 72 h in standard co-culture (LC + MS-5) or 3D SAPH (*n* = 3 for LC + MS-5; *n* = 4 for Hydrogel). (**B**) Representative phase-contrast images, and (**C**) efficiency as percentage of control (*n* = 3) of colony formation in AML patient samples (#522 and #694) 14 days following treatment with 0.02% DMSO vehicle (Control), 5 µM Atorvastatin, or 500 nM Salinomycin.

## Data Availability

The data presented in this study are available on request from the corresponding author.

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
