# Peer review of "Hydrogel-Based Pre-Clinical Evaluation of Repurposed FDA-Approved Drugs for AML"

_ijms, 2023, doi:10.3390/ijms24044235_

Round 1

Reviewer 1 Report

Self-assembling peptide hydrogels (SAPH) are a highly popular biomaterial due to their biocompatibility, design, and structural properties, which make them a suitable candidate for a specific 3D microenvironment with the potential for cell encapsulation and drug delivery. In this manuscript, the authors developed a 3D model of the bone marrow niche to screen repurposed FDA-approved drugs using self-assembling peptide hydrogels (SAPH). In addition, the stiffness of the SAPH is optimized to facilitate the growth of colonies. In this context, three FDA-approved drug candidates are tested against the THP-1 cell line and primary mAF9 cells, and half of the maximum effective concentration (EC50) is used to evaluate the sensitivity of the three drugs Salinomycin, Vidofludimus and Atorvastatin in the peptide hydrogel models. This paper is well structured, and the results presented are of significance to the scientific community.

I would recommend it for publication after addressing the following comments.

Major concerns:

1.    A high peptide concentration results in a stiffer gel, which ultimately affects the cell structure and colonies. Quantitative data is needed to support this statement. Also, I would like to see the difference in cell structure and colonies in relation to high peptide concentrations.

2.   A schematic representation of the 3D hydrogel and cell encapsulation is needed in the Results and Discussion section.

3.    What was the control in Figures 3 and 4? I suggest adding a bar graph with the control experiment showing the survival percentage.

4.    What are the rheological properties of the 3D hydrogel that were considered? An explanation is needed in the Materials and Methods section.

5.    The effect of Atorvastatin and Salinomycin on the viability of AML patient samples in self-assembling peptide hydrogel (SAPH) is discussed in Section 2.5. How many measurements were performed? More precise quantification is required in Figure 6.

6. The effect of three drugs, Salinomycin, Vidofludimus, and Atorvastatin, on THP-1 and mAF9 cells in liquid culture shows a more pronounced trend at 48 hours as a function of drug concentration. What mechanism enhances this effect? Also, write the viability of the cells in numbers in the text.

Minor comments:

1.    CO2 to CO2, make it consistent throughout the manuscript.

2.    The resolution of Figure 4, Figure 5B, and Figure 6 needs to be improved.

3.    Figure 1A, x-axis, mg/ml to mg/mL

Reviewer 2 Report

In this manuscript James et al. examine the advantages of using synthetic, self-assembling peptide hydrogel (SAPH), as a 3D model representing bone marrow niche to screen repurposed (FDA approved) candidate drugs for AML therapy. Although the topic of this investigation is really important for the field, the study needs some improvements in order to be suitable to be published in this Journal.

Major concerns:

Introduction:

Candidate repurposed drugs that were examined in this study should also be mentioned and their choice should be justified in the Introduction section.

Hydrogel advantages over other static 3D models (for example, Matrigel) were discussed in the Introduction. However, comparisons between hydrogel models and dynamic 3D in vitro models (bioreactors) should also be made.

 Results:

It should be explained why only U937 cells were used to optimise the concentration of peptide in the hydrogel model, as those cells have not been used for further experiments.

Name of 2.2. subsection should be corrected as provided data discusses maintenance and survival of primary marine leukemia cells not only in unmodified but also in modified SAPH.

It seems that in subsection 2.2. standard (1x) and reduced growth factor (0.5x) liquid culture medias were incorrectly denoted (either in text, or in Figure 2).

Authors should provide explanation why only those two particular cell lines (TPH-1 and mAF9) were tested for sensitivity towards Salinomycin, Vidofludimus and Atorvastatin, although additional four cell lines were tested for grow and colonies formation in hydrogel model?

Confidence intervals for EC50 data (Figure 3B, Figure 4B) should be provided.

Group of AML patient samples is very small (only two patient samples); in addition, patient diagnosis and treatments are not provided.

Statistical analysis is lacking.

 Materials and Methods:

Method of EC50 calculation is not described in Materials and Methods.

Method of mAF9 cells generation should be explained in Materials and Methods.

Authors declare that “all batches of peptide were analysed for similarity in rheological properties prior to use to ensure comparability of results” (lines 312-314). For repeatability purposes this data should also be provided in Materials and Methods.

Minor concerns:

Quality of figures (Figure 1, 2, 4, 5, 6) should be improved.

Formatting should be corrected (line 358).

Round 2

Reviewer 1 Report

The authors have adequately addressed my concerns with the original draft and have significantly improved the quality of the paper as a result. The manuscript is now suitable for publication in IJMS.

Reviewer 2 Report

Authors have revised the manuscript thoroughly and, in my opinion, it is now acceptable for publication.